# Activation of Adrenoceptor Alpha-2 (ADRA2A) Promotes Chemosensitization to Carboplatin in Ovarian Cancer Cell Lines

Haya Albanna [†] , Alesia Gjoni [†], Danielle Robinette, Gerardo Rodriguez, Lora Djambov, Margaret E. Olson * and Peter C. Hart *

College of Science, Health and Pharmacy, Roosevelt University, 1400 N Roosevelt Blvd, Schaumburg, IL 60173, USA; halbanna@mail.roosevelt.edu (H.A.); agjoni@mail.roosevelt.edu (A.G.); drobinette@mail.roosevelt.edu (D.R.); grodriguez24@mail.roosevelt.edu (G.R.); ldjambov@mail.roosevelt.edu (L.D.)

* Correspondence: mkurbanov01@roosevelt.edu (M.E.O.); phart02@roosevelt.edu (P.C.H.); Tel.: +1-(847)-330-4542 (M.E.O.); +1-(847)-330-4505 (P.C.H.)

† These authors contributed equally to this work.

**Abstract:** Recurrence of ovarian cancer (OvCa) following surgery and standard carboplatin/paclitaxel first-line therapy signifies poor median progression-free survival (<24 months) in the majority of patients with OvCa. The current study utilized unbiased high-throughput screening (HTS) to evaluate an FDA-approved compound library for drugs that could be repurposed to improve OvCa sensitivity to carboplatin. The initial screen revealed six compounds with agonistic activity for the adrenoceptor alpha-2a (ADRA2A). These findings were validated in multiple OvCa cell lines (TYKnu, CAOV3, OVCAR8) using three ADRA2A agonists (xylazine, dexmedetomidine, and clonidine) and two independent viability assays. In all the experiments, these compounds enhanced the cytotoxicity of carboplatin treatment. Genetic overexpression of ADRA2A was also sufficient to reduce cell viability and increase carboplatin sensitivity. Taken together, these data indicate that ADRA2A activation may promote chemosensitivity in OvCa, which could be targeted by widely used medications currently indicated for other disease states.

**Keywords:** ovarian cancer; OvCa; carboplatin resistance; ADRA2A; adrenoceptor alpha-2a

## 1. Introduction

Ovarian cancer (OvCa) commonly presents at advanced stages and has a poor prognosis, with an overall 5-year survival rate of 50% across all subtypes and stages at diagnosis [1]. The standard of care for high-grade serous ovarian carcinoma (HGSOC), the most prevalent subtype of OvCa, includes cytoreductive debulking surgery and chemotherapy consisting of a platinum agent such as carboplatin (CP) and a taxane such as paclitaxel [2]. Following active transport, CP promotes DNA lesions that include direct platinum–DNA adducts as well as inter- and intra-strand crosslinking that ultimately result in single- and double-strand breaks, inducing apoptosis [3]. In a clinical context, the use of CP/paclitaxel (CarboTaxol) allows for the ablation of unresectable or visually undetectable tumor cells to prevent recurrence of the disease.

While the majority of patients initially respond to CP therapy, the majority will relapse and inevitably succumb to their disease [2,4,5]. Multiple mechanisms have been attributed to CP treatment failure, including dysfunctional transport, compensatory DNA repair, and the upregulation of pro-survival and anti-apoptotic pathways [6]. The use of paclitaxel in combination to impede progression through the M phase of cells that evade apoptosis despite DNA adduct accumulation by carboplatin may be central to its ability to enhance tumor cell ablation following surgery and to improve the initial therapeutic response. Conceivably, drugs targeting other processes that evade apoptosis or promote survival may

complement the current therapeutic approaches [7]. In line with this, paclitaxel has been used in conjunction with carboplatin as first-line therapy to improve survival in patients with advanced HGSOC due to its inhibition of microtubule disassembly and subsequent cell cycle arrest to further enhance the apoptosis caused by CP-dependent DNA damage [8]. Emerging molecular targets have been identified that may delay recurrence following first-line treatment in OvCa, such as VEGF (e.g., bevacizumab) and PARP (e.g., olaparib), which have provided new therapeutic strategies to improve progression-free survival [9]. In the case of PARP inhibitors, suppression of nucleotide excision repair may further enhance the efficacy of CP by preventing the removal of intrastrand crosslinking to promote apoptosis [6,10]; however, PARylation is critical in regulating numerous processes involved in DNA repair and replication that may be involved in preventing the proliferation of CP-treated cells [11]. While promising, in both cases of VEGF and PARP inhibitors, it has been observed that treatment failure may still occur due to resistance that develops in response to therapy [12,13]. Thus, it is critical to continue expanding the repertoire of nonredundant treatment options available to clinicians that may facilitate personalized treatment plans to overcome any acquired drug resistance and to prolong patient survival.

To expedite the implementation of novel targetable mechanisms for drug therapies, a drug repurposing paradigm is often used in the drug discovery process for many disease states. This approach allows researchers to proceed more rapidly to clinical trials to determine efficacy and has facilitated the common use of multiple drugs as mainstays in current cancer therapy, such as tamoxifen in breast cancer [14,15]. Many of the drugs repositioned for off-label use in these contexts have been evaluated as a result of retrospective cohort analyses that have identified associations between disease onset or progression and medications taken for other indications [14]. Although these retrospective associations have, in many cases, facilitated preclinical experimentation and the pursuit of clinical trials to determine efficacy, their application in new clinical contexts is not always successful [14]. An alternative strategy for drug repurposing has been through the prospective preclinical assessment of drugs already on the market. In particular, in vitro high-throughput screening (HTS) using curated compound libraries to evaluate thousands of compounds rapidly [16] allows for lead-compound identification and in vivo testing prior to clinical application. Indeed, a recent study using a synthetic lethality HTS approach identified FDA-approved drugs that may improve the response to ATP8B-dependent cisplatin resistances in OvCa [17], indicating the potential use of these compounds (e.g., tranilast) in ATP8B-overexpressing ovarian tumors. Using a similar unbiased strategy, the current study aimed to enhance the cytotoxicity of carboplatin to prevent treatment resistance by utilizing an FDA-approved compound library.

## 2. Materials and Methods

### 2.1. Reagents

High-throughput screening (HTS) was performed using the DiscoveryProbe L1021 FDA-approved drug library (APExBIO L1021). Carboplatin (#c2538), xylazine (#x1126), dexmedetomidine (#sml0956), and clonidine (#c7897) were purchased from Sigma-Aldrich.

### 2.2. Cell Lines

The OvCa cell lines TYKnu, CAOV3, and OVCAR8 were a gift from Dr. Ernst Lengyel and Dr. Iris Romero of the University of Chicago Section of Gynecologic Oncology (Chicago, IL, USA). All cell lines were passaged at least 3 times prior to their use in experiments. Cell lines were banked in liquid nitrogen and were confirmed to be mycoplasma-free by the donating investigators as described previously [18]. Viability was confirmed to be 98% or higher (using trypan exclusions with Countess II, ThermoFisher, Waltham MA, USA) prior to counting and seeding for the HTS as well as the primary and secondary assays. All three cell lines were maintained in RPMI1640 with L-glutamine (VWR) supplemented with 10% FB-Essence (VWR), 1% penicillin/streptomycin (VWR), 1% MEM vitamins (VWR), and 1% nonessential amino acids (VWR).

### 2.3. High-Throughput Screening (HTS)

The initial high-throughput screening (HTS) was performed using a 96-well format with the DiscoveryProbe L1021 FDA-approved drug library (ApexBio, Houston, TX, USA). The compound library was reconstituted to 2 mM (in DMSO) so that the addition of 5 μL of each drug would yield 50 μM per drug per well. A total of 88 compounds were tested per plate, with 1584 compounds tested. For each plate, internal controls included 4 untreated samples and 4 samples treated only with the $IC_{50}$ of carboplatin. Due to their relatively high resistance to carboplatin, as indicated by the highest $IC_{50}$ of the cell lines tested, OVCAR8 cells were used in the pilot HTS assay. Cells were seeded at $3.0 \times 10^4$/well and allowed to grow for 24 h prior to treatment with carboplatin (at $IC_{50}$) ± the L1021 drug library for 48 h. After 48 h of treatment, the MTT assay was performed (as described below) to determine the effect on cell viability. Absorbance measurements were then processed in which the absorbance of each drug with carboplatin was compared to the positive carboplatin control (i.e., the $IC_{50}$ of carboplatin alone). Any well which was observed to increase the cytotoxicity of carboplatin by 80% or higher was considered a potential lead (marked as red; Figure 1E and Supplementary Table S1). These statistics were then cross-referenced per well per plate to identify compounds of interest for subsequent primary (MTT viability) and secondary (colony formation) confirmatory assays.

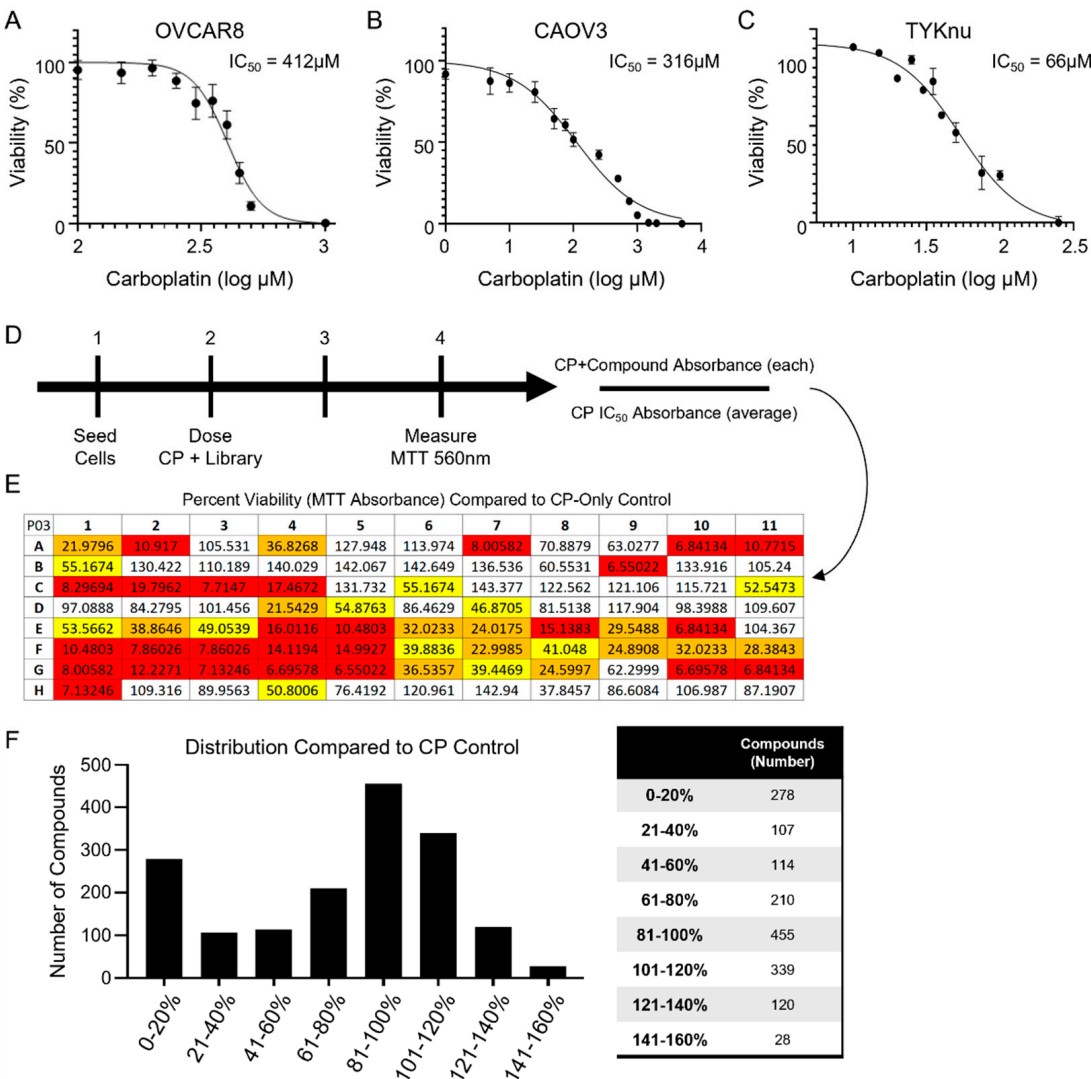

**Figure 1.** Determination of carboplatin resistance in OvCa cell lines. Cells were plated and allowed to adhere for 24 h prior to treatments. Cells were treated with 12 escalating doses of carboplatin (0–5000 μM)

for 48 h ((**A**) OVCAR8, (**B**) CAOV3, and (**C**) TYKnu cell lines). N = 4 for each data point. (**D**) Experimental timeline for treating cells with carboplatin at its determined IC$_{50}$ along with the DiscoveryProbe L1021 FDA-approved drug library (ApexBio) (88 agents per plate at 50 μM final concentration). (**E**) MTT absorbance was normalized to the carboplatin-only group to determine the impact on cell death. Numbers presented refer to the percentage (%) of carboplatin-only control. Wells marked in red had an absorbance of <20% when divided by the average absorbance of the carboplatin-only controls for each plate. (**F**) The frequency distribution of compounds + carboplatin binned in 20% increments, with 100% representing having an absorbance equal to the carboplatin-only controls. The number of compounds identified in each category is shown on the right.

### 2.4. Primary Assay—MTT Viability Assay

The thiazolyl blue tetrazolium bromide (MTT) colorimetric metabolism assay was used to determine cell viability (GoldBio, St. Louis, MO, USA; #T-030). Briefly, cells were seeded at $3.0 \times 10^4$/well by dispensing 100 μL of a $3.0 \times 10^5$/mL suspension of cells into a 96-well plate (Pipette.com, San Diego, CA, USA) and allowed to adhere for 24 h prior to treatment. Media was aspirated and fresh growth media was added so that the addition of carboplatin ± secondary compounds would result in a final volume of 200 μL. Carboplatin was diluted fresh from 10 mM stocks (never previously thawed) to working stocks to allow for 5 μL of drug to result in the desired concentrations (ranging from 10–5000 μM). In the context of the HTS, only the IC$_{50}$ calculated for OVCAR8 was used in combination with each drug. In the context of the MTT confirmatory experiments, a dose range was used to reestablish the IC$_{50}$ per plate (carboplatin alone) and to determine IC$_{50}$ values when used in combination with each alpha-2 agonist (set to 50 μM and 100 μM per agonist). Multichannel pipettes were used to limit variability between wells. After 48 h of treatment, 12 mM MTT was added to a final concentration of 300 μM and allowed to incubate for 1.5 h (OVCAR8) or 2 h (CAOV3 or TYKnu). After incubation, the plates were decanted and patted dry, and MTT-reduced formazan crystals were suspended in 200 μL DMSO. The absorbance was then measured at 560 nm using SpectraMax ABS (MolecularDevices, San Jose, CA, USA). The absorbance values were exported via Microsoft Excel and analyzed using GraphPad Prism (described below).

### 2.5. Secondary Assay—Colony-Formation Clonogenicity Assay

As a secondary assay to confirm the impact of each drug combination on cell chemoresistance, a colony formation (clonogenicity) assay was performed. Cells were seeded at 500 ($0.5 \times 10^3$; OVCAR8) or 1000 ($1.0 \times 10^3$; CAOV3, TYKnu) cells per well of a 6-well plate (VWR, Radnor, PA, USA) and allowed to recover for 24 h. After 24 h, drug treatments were initiated. At days 5 and 7, the drug treatments were refreshed, with a full media change performed on day 5. On day 10, the cells were washed with 1× PBS, fixed using 4% paraformaldehyde for 15 min, washed again with 1× PBS, then stained using crystal violet (Sigma-Aldrich, St. Louis, MO, USA, #61135) for 30 min. The crystal violet was then moved to a waste container, and the cells were rinsed with water until the stain was washed away. Plates were then imaged using ChemiDoc XR+ (Bio-Rad, Hercules, CA, USA) while colonies were counted manually using ImageJ/FIJI (NIH, Bethesda, MD, USA), and statistical analyses were performed using GraphPad Prism.

### 2.6. Plasmid Transfection

To overexpress adrenoceptor alpha-a2, transfection was performed using pmEGFP-ADRA2A on the pcDNA3.1 backbone (Addgene, Watertown, MA, USA; plasmid #190753). pmEGFP-ADRA2A was a gift from Dave Piston of Washington University. OVCAR8 cells were seeded at $4.0 \times 10^6$ per 10 cm plate and allowed to proliferate for 24 h prior to transfection. Cells were transfected using Lipofectamine 2000 (Invitrogen, Waltham, MA, USA) according to the manufacturer's instructions. At 24 h post transfection, cells were seeded into either 96-well or 6-well plates for the MTT or colony formation assays,

respectively. After 48 h post transfection, treatments were initiated as described for the primary and secondary assays. For quantitative real-time PCR, the reaction was scaled down to a 6-well format with $1\times$ reaction per well.

### 2.7. Quantitative Real-Time Polymerase Chain Reactions

The expression of ADRA2A following transfection was confirmed by quantitative real-time PCR (qRT-PCR). RNA isolation was performed using Trizol (Invitrogen) with chloroform extraction. Complementary DNA (cDNA) synthesis was performed using the High-Capacity cDNA Reverse Transcription Kit (Applied Biosystems, Waltham, MA, USA). qRT-PCR was performed using the POWRUP SYBR Master Mix (Life Technologies, Carlsbad, CA, USA) according to the manufacturer's instructions. Primers were designed using PrimerBank (NP_000672, #194353969c2), and 25 nmole quantities of DNA oligonucleotides were purchased from IDT (Newark, NJ, USA; primers were prepared as LabReady, normalized to 100 μM in IDTE pH 8.0). Forward primer: AGA AGT GGT ACG TCA TCT CGT; Reverse primer: CGC TTG GCG ATC TGG TAG A.

### 2.8. TCGA Assessment of ADRA2A Expression in Ovarian Cancer

Expression of ADRA2A in ovarian cancer was determined using the Gene Expression Profiling Interactive Analysis (GEPIA) database (GEPIA.cancer-pku.cn). The "OV" (ovarian cancer) dataset from The Cancer Genome Atlas (TCGA) was selected with a $\log_2$FC cutoff of 1 and a *p*-value cutoff of 0.01. The data are presented on a log scale.

### 2.9. Statistical Analysis

Data was analyzed using Graphpad Prism 10 (Graphpad, La Jolla, CA, USA) by 1-way ANOVA with the post hoc Tukey t-test or Student's unpaired t-test as appropriate. For the $IC_{50}$ calculations, XY data tables and graphs were used. Absorbance measurements from the MTT assays were imported into Prism. The data were then transformed to $\log(x)$ and normalized as a percentage within each group. The transformed/normalized data were then modeled using nonlinear regression (dose response: log(inhibitor) with four parameters, with constraints for "top" (100%) and "bottom" (0%)). Representative curve fit and calculated $IC_{50}$ values are presented for each experiment. All experiments were repeated at least once.

## 3. Results

### 3.1. High-Throughput Screening of an FDA-Approved Drug Library Identifies Adrenoceptor Alpha-2 (ADRA2A) as a Potential Target for Enhancing Carboplatin Treatment

To identify novel compounds that could enhance the cytotoxicity of carboplatin and, thus, possibly reduce resistance to the current standard of care, three cell lines representative of HGSOC were assessed for their potential resistance to carboplatin (CP). Treatment of TYKnu, CAOV3, and OVCAR8 cells with increasing doses of carboplatin for 48 h revealed significant resistance of OVCAR8 cells compared with either CAOV3 or TYKnu, with an $IC_{50}$ value of 412 μM (Figure 1A) compared with either 316 μM (Figure 1B) or 66 μM (Figure 1C), respectively, as determined by the MTT cell viability assay. OV-CAR8 cells were therefore selected for high-throughput screening (HTS) utilizing an FDA-approved drug library (APExBIO L1021). Cells were treated with the relative $IC_{50}$ of CP (400 μM) $\pm$ compound library (50 μM) for 48 h prior to the MTT assay (Figure 1D; example plate analysis shown in Figure 1E). Of the 1651 compounds tested, 278 demonstrated a marked enhancement in cell death that increased the cytotoxicity of CP by more than 80% compared with the carboplatin-only controls (Figure 1F and Supplementary Table S1).

Of the 278 compounds observed to have potential synergy with CP, 4 were identified to be agonists of the ADRA2A receptor, including xylazine, tizanidine, medetomidine, and dexmedetomidine (Figure 2A). Two agents with mixed agonism of ADRA2A and significant $I_1$-imidazoline receptor activity were also identified: moxonidine and rilmenidine (Figure 2A). This trend indicated that ADRA2A could be an important molecular target that

may regulate chemosensitivity in OvCa tumor cells. Evaluation of ADRA2A expression in patient tumors from the TCGA database revealed that ADRA2A is significantly repressed in OvCa tumors (Figure 2B), further supporting the hypothesis that inhibition of its signaling may be important in tumor development or progression.

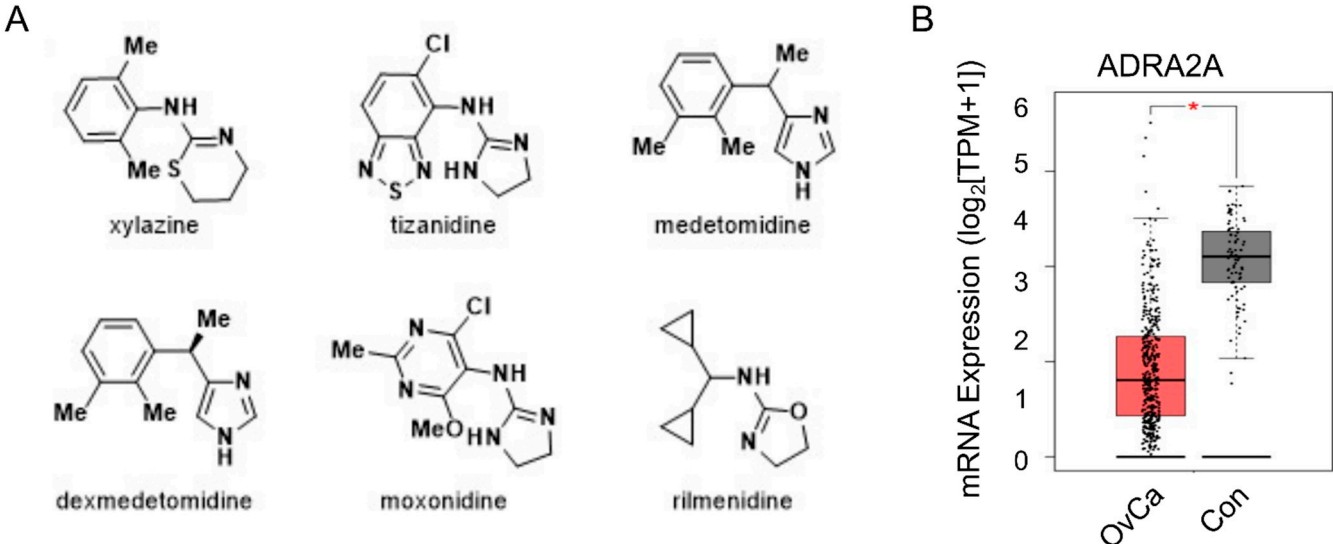

**Figure 2.** Identification of ADRA2A as a potential therapeutic target in OvCa. (**A**) High-throughput screening (HTS) of carboplatin + the FDA library identified several alpha-2 adrenoceptor (ADRA2A) agonists that increased cytotoxicity by >80% in combination with carboplatin; these included xylazine, tizanidine, medetomidine, and dexmedetomidine. Drugs with partial ADRA2A agonism, namely moxonidine and rilmenidine, were also identified in the HTS. (**B**) Gene (mRNA) expression of ADRA2A in TCGA clinical data (GEPIA database) comparing control tissue (gray) to ovarian tumor tissue (pink). * $p < 0.01$.

### 3.2. Activation of ADRA2A Using Pharmacologic and Genetic Manipulation Enhances Carboplatin Toxicity in Viability and Clonogenicity Assays

To validate the findings from the HTS in OVCAR8 cells, TYKnu, CAOV3, and OVCAR8 cells were treated with increasing doses of carboplatin ± the ADRA2A agonists xylazine, dexmedetomidine, or clonidine. While not observed in the HTS, clonidine was chosen due to its widespread clinical use for hypertension, relatively high specificity, and low toxicologic profile. Primary confirmatory MTT assays in these cell lines demonstrated consistent reductions in the $IC_{50}$ of carboplatin compared with the carboplatin-only groups when 50 or 100 μM of any of the ADRA2A agonists was co-administered (Figure 3A–C). This effect was largely dose-dependent and was consistent across the three cell lines tested (Figure 3D). To determine whether this effect was assay-specific, a secondary colony-formation clonogenicity assay was performed using the approximate $IC_{50}$ value of CP (1 μM) in combination with each compound. Extended treatment of colonies for 10d with a lower dose (25 μM) of each compound suppressed colony formation, an effect that was particularly significant with xylazine and dexmedetomidine (Figure 4A–C). Clonidine appeared to have weak activity in this assay at the test dose and reached significance in CAOV3 cells (Figure 4B) but did not meet statistical significance in TYKnu (Figure 4A) or OVCAR8 (Figure 4C) cells, despite showing a similar trend [$p < 0.09$].

To further evaluate our findings using the pharmacologic activation of ADRA2A, genetic upregulation of ADRA2A in the most resistant cell line (OVCAR8) was performed (Figure 5A). Following the ectopic upregulation of ADRA2A, cells were subjected to either MTT or colony formation assays. ADRA2A overexpression in OVCAR8 cells strikingly reduced the $IC_{50}$ of carboplatin needed in these cells (Figure 5B). Consistently, ADRA2A expression induced an overall inhibition of colony formation in OVCAR8 cells and promoted

a further reduction in colonies compared with the carboplatin-only group (Figure 5C). In line with the enhancement in carboplatin sensitivity observed using xylazine, dexmedetomidine, and clonidine, these findings suggest that ADRA2A activation and the consequent inhibition of cyclic AMP (cAMP) production may downregulate processes necessary for tumor cells to escape carboplatin-induced cytotoxicity. Thus, taken together, these data further support the idea that the molecular activity of ADRA2A may suppress the resistance of OvCa tumor cells to carboplatin.

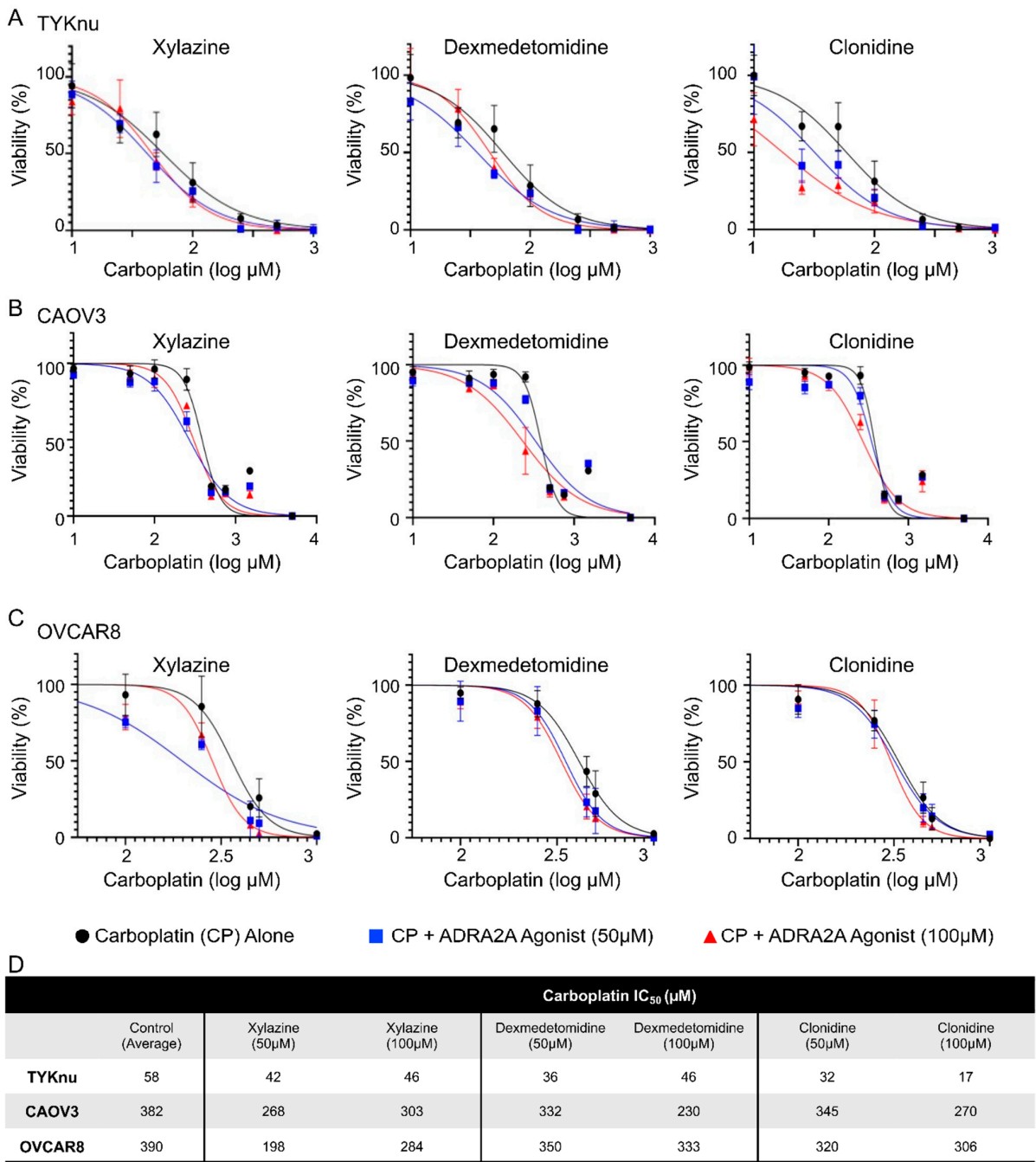

**Figure 3.** Validation of ADRA2A agonists in enhancing carboplatin-dependent cytotoxicity. Cells were plated and allowed to adhere for 24 h prior to treatments. Cells were treated with escalating doses of carboplatin (0–5000 µM) ± 50 or 100 µM of the indicated ADRA2A agonist for 48 h ((**A**) TYKnu, (**B**) CAOV3, (**C**) OVCAR8). N = 4 for each data point. (**D**) Carboplatin $IC_{50}$ values for carboplatin ± ADRA2A agonists (xylazine, dexmedetomidine, clonidine) for each cell line.

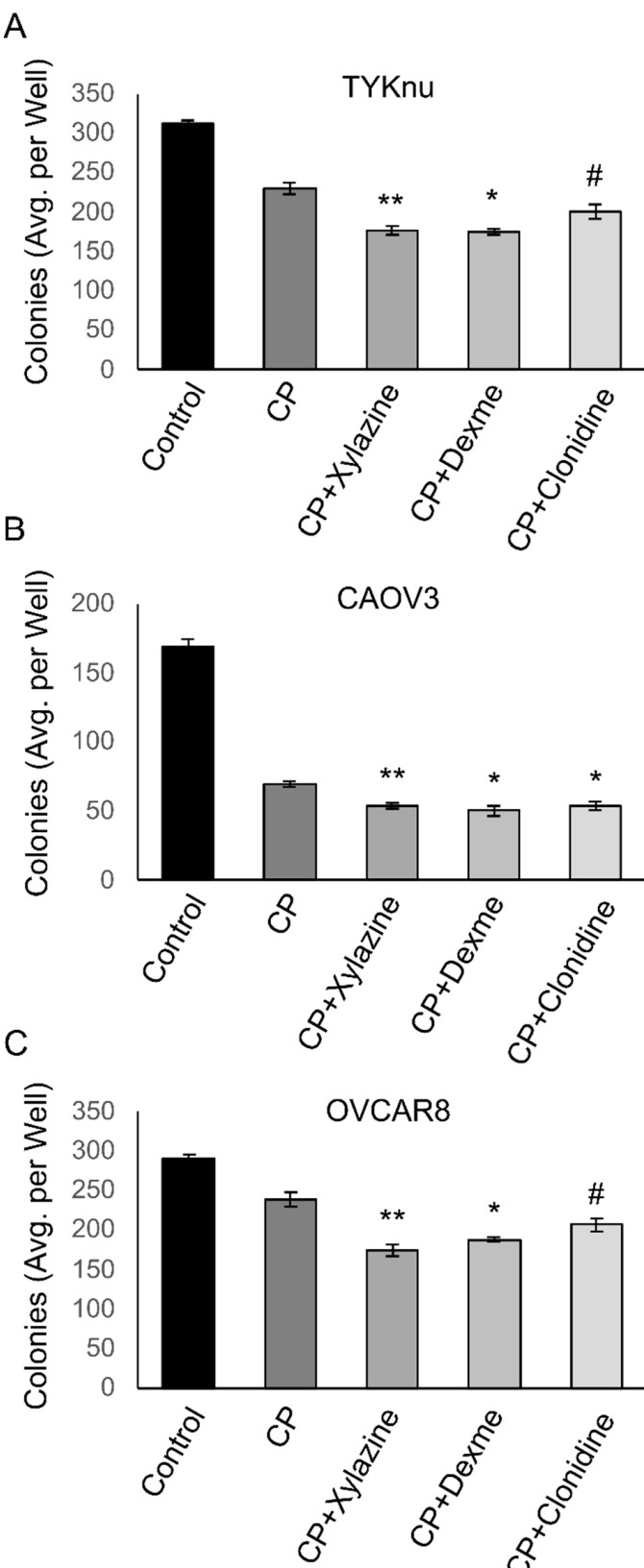

**Figure 4.** ADRA2A activation further enhances the inhibition of clonogenicity by carboplatin. Cells were allowed to adhere for 24 h prior to treatments. Cells were treated with 1 μM carboplatin ± 25 μM of the indicated ADRA2A agonist for 10 days. ((**A**) TYKnu, (**B**) CAOV3, (**C**) OVCAR8). N = 3 for each condition. * $p < 0.05$; ** $p < 0.01$; # $p < 0.07$. Dexme = dexmedetomidine; CP = carboplatin.

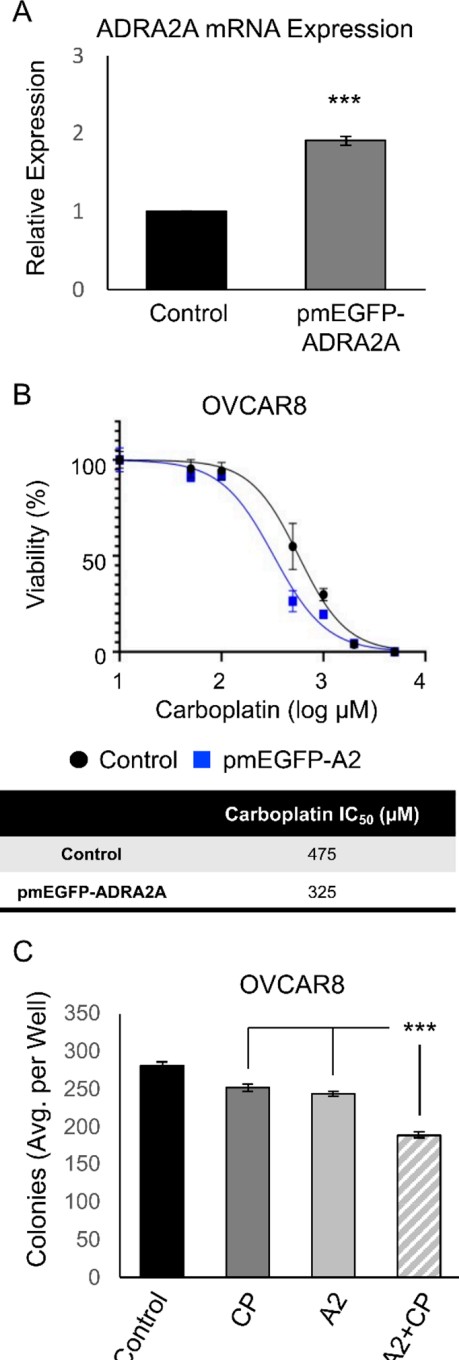

**Figure 5.** Overexpression of ADRA2A promotes chemosensitivity to carboplatin. (**A**) Expression of ADRA2A mRNA at 48 h following the transfection of OVCAR8 cells with pmEGFP-ADRA2A (N = 3). (**B**) MTT viability assay of OVCAR8 cells ± pmEGFP-ADRA2A treated with an escalating dose curve of carboplatin (0–5000 μM) for 48 h (N = 6). (**C**) Colony formation assay of OVCAR8 cells ± pmEGFP-ADRA2A [A2] treated with or without carboplatin (1 μM) for 10 days (N = 6). *** $p < 0.005$.

## 4. Discussion

In summary, an unbiased HTS cytotoxicity assay identified that several modulators of ADRA2A activity could enhance the sensitivity of OvCa cell lines to carboplatin. The initial HTS data was confirmed using the compounds xylazine, dexmedetomidine, and clonidine sourced from a different manufacturer for validation in both primary (MTT viability) and

secondary (colony formation) assays. These compounds demonstrated a reduction in the $IC_{50}$ of CP in all three cell lines tested by the MTT viability assay. When combined with CP, xylazine and dexmedetomidine also showed a marked reduction in the number of colonies formed by all three cell lines. While the use of the three compounds in these assays was largely consistent, clonidine did not reach the same statistical significance compared with xylazine and dexmedetomidine in the colony formation assay in two of the cell lines. This could be in part due to differences in its potency, a difference in specificity for the three $\alpha2$ subunits (A–C), or its mixed activity on the $I_1$ imidazoline receptor; however, in the case of the latter, dexmedetomidine has also been shown to be pharmacologically active in a nonadrenergic $I_1$ receptor manner [19]. In addition to assessing the impact of $\alpha2$ B and C subunits on OvCa chemoresistance, further evaluation of the $I_1$ receptor may indicate its usefulness as a target in OvCa and may indicate whether increased specificity for ADRA2A may alter its potential anticancer effects; this may be best accomplished using moxonidine and rilmenidine due to their higher selectivity for $I_1$ compared with clonidine [19]. Regardless, the data presented herein indicate the potential for ADRA2A-modulating antihypertensives (clonidine) or sedative analgesics (dexmedetomidine), which are relatively well tolerated and have well defined pharmacokinetics, to serve as an adjunct to improve the efficacy of carboplatin. This notion was further supported by the genetic upregulation of ADRA2A, which demonstrated a striking sensitization of the highly chemoresistant OVCAR8 cell line to carboplatin.

While the present study did not delineate the exact mechanisms underlying the potential CP sensitization, multiple downstream effectors of ADRA2A have been well described in multiple cancers. ADRA2A belongs to the adrenoceptor alpha-2 family of GPCRs coupled to the inhibitory G-alpha subunit (Gi) that suppresses adenylyl cyclase activity, thereby suppressing the cyclization of AMP to cAMP [20]. cAMP is a key second messenger that impacts multiple intracellular pathways, canonically regulating protein kinase A (PKA) and cAMP-response-element-binding protein (CREB) activity, but also calcium-dependent signaling and mitogenic pathways such as MAPK [21]. While the functional role of cAMP varies by tissue, in OvCa, cAMP signaling has been associated with enhanced proliferation and the suppression of pro-apoptotic signals, frequently associated with the activity of PKA or CREB1 (reviewed in [22]). Notably, PKA activation has been observed to promote platinum resistance in ovarian and other cancers [22–24], and CREB1 inhibition potently sensitized OvCa tumor cells to cisplatin [25]. In a recent study, ADRA2A overexpression was observed to suppress proliferation and to promote apoptosis through the repression of PI3K/AKT/mTOR signaling in cervical cancer [26], another signaling axis associated with platinum resistance in several contexts [6]; however, to date, no work has demonstrated the impact of ADRA2A activity in OvCa development or treatment specifically. Further evaluation is needed given the many potential downstream pathways that may be attributed to the effects of ADRA2A activation observed in the current work.

To date, sparse data exist that show the clinical impact of the role of alpha adrenoceptor activity in tumor development and progression. Although antihypertensives, including beta blockers, have not been associated with OvCa patient survival [27], the effects of alpha adrenoceptor activation have yet to be evaluated in this clinical context. As clonidine has been used to mitigate symptoms in OvCa patients following bilateral oophorectomy [28], it may be possible to retrospectively analyze whether the inclusion of this drug may delay the time to recurrence in these patients. To increase the clinical relevancy of our findings, an evaluation of whether ADRA2A activity is relevant in a more physiologic model system that includes components of the OvCa tumor microenvironment (TME) (as performed in a recent study [18]) is required to improve the translatability of these findings, considering that cancer-associated fibroblasts in the TME are thought to significantly contribute to platinum resistance in multiple contexts [29,30]. This may be especially important given recent findings observing that dexmedetomidine induced IL-6 secretion from stromal stellate cells and pro-tumorigenic signaling in hepatocellular cancer cells [31], although whether this is relevant to that distinct TME is to be determined. Moreover, the beta-

adrenergic antagonist propranolol has been shown to suppress tumor growth by inhibiting tumor angiogenesis and promoting T-cell recruitment in sarcoma [32], and the nonselective blockade of beta-adrenoceptors in prostate and pancreatic cancer in vivo suppressed tumor growth [33]. Additionally, beta-adrenergic activation has been associated with remodeling of the extracellular matrix to promote invasive phenotypes in breast cancer [34], which may be, in part, due to the potential cAMP-mediated epithelial–mesenchymal transition (EMT) [35,36]; however, whether ADRA2A inhibition of this signaling pathway results in suppression of the EMT is still to be determined. Those results and ours here suggest a complex relationship among adrenergic receptor subtypes and the processes involved in tumor development in a context-dependent manner. Further evaluation of the effects of specific adrenoceptor isoforms in tumor and stromal cells within the TME may provide a clearer understanding of the possible therapeutic value of targeting noradrenergic signaling. Lastly, in vivo data utilizing an orthotopic approach combined with a CarboTaxol regimen will ultimately determine whether the agents identified here would be suitable for inclusion in the therapy for advanced HGSOC.

## 5. Conclusions

Our HTS findings indicate that a number of currently approved compounds may enhance the chemosensitivity of OvCa cells to CP. Of these, ADRA2A-targeted compounds were highly represented in the HTS. Validation of several of these compounds supports the hypothesis that the activation of ADRA2A may reduce the chemoresistance of OvCa tumor cells and improve the response to CP. These pharmacologic data were in agreement with the effect of genetically upregulating ADRA2A, which resulted in the sensitization of chemoresistant OvCa cells to CP.

**Supplementary Materials:** The following supporting information can be downloaded at: https://www.mdpi.com/article/10.3390/cimb45120598/s1, Table S1: Viability of OvCa cells in response to carboplatin and the FDA -approved compound library. Percentage of viable cells treated with carboplatin + the FDA library relative to the carboplatin-only control.

**Author Contributions:** The manuscript was written through contributions from all authors. M.E.O. and P.C.H. led the conceptual and experimental design as well as the development of the methodology. H.A., A.G., D.R., G.R., L.D. and P.C.H. performed and analyzed the experiments, and contributed to the manuscript writing and editing. M.E.O. contributed to the manuscript writing, editing, and revision. M.E.O. and P.C.H. contributed equally to the supervision of the study. All authors have read and agreed to the published version of the manuscript.

**Funding:** This work was supported, in part, by the American Association of Colleges of Pharmacy (New Investigator Award to PCH) and the Howard Hughes Medical Institute (Inclusive Excellence Grant to Roosevelt University).

**Institutional Review Board Statement:** Not applicable.

**Informed Consent Statement:** Not applicable.

**Data Availability Statement:** Original data is available from the corresponding authors upon request.

**Acknowledgments:** The authors thank Melissa Hogan and Lawrence A. Potempa for their support in acquisition of funds for the reagents and equipment used to procure data in this manuscript.

**Conflicts of Interest:** There are no conflict of interest to report.

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
