# Peer review of "Activation of Adrenoceptor Alpha-2 (ADRA2A) Promotes Chemosensitization to Carboplatin in Ovarian Cancer Cell Lines"

_cimb, doi:10.3390/cimb45120598_

Round 1
Reviewer 1 Report
Comments and Suggestions for Authors
I have read the manuscript and it well researched and structured by the authors. I would like to endorse the manuscript for publication.
Comments on the Quality of English LanguageThe quality of the english is very good and readeble.
Author Response
The Authors thank the Reviewer for their time in reviewing our manuscript, and we look forward to sharing our findings with the field. Please see attachment for the full cover letter including response to all Reviewers' critiques and the revised version of the manuscript.

Reviewer 2 Report
Comments and Suggestions for Authors
The authors performed a combinational screening with carboplatin and compounds from the DiscoveryProbe L1021 FDA-approved Drug library in carboplatin-resistant ovarian carcinoma cell lines. The readout was by MTT and by Colony Formation Clonogenicity Assay. The mechanism of the identified compounds was validated by overexpression of the presumed target, the adrenoreceptor alpha-a2.
The study was performed state of the art and results are well documented and support the conclusions made by the authors.
It would be great if population data would also support this target. Whereas exposure of humans to Xylazine and dexmedetomidine is short short periods, clonidine can be used for chronic treatment e.g. for hypertension. Is there any indication that patients with long-term use of this drug have a lower incidence of ovarian carcinoma or a better response to carboplatin or a better overall prognosis of ovarian carcinoma? Would it be possible to include such an information in the article?
Author Response
The Authors thank the Reviewer for this comment. It would be ideal to retrospectively analyze patient data to determine if there is an association with antihypertensives, specifically clonidine, and incidence or mortality of ovarian cancer as well as time to recurrence. As pointed out by the reviewer, both xylazine and dexmedetomidine have fairly limited use and are only clinically used acutely, so it may be difficult to adequately assess the clinical impact of either drug in this case.
The only study to date that we are aware of showed no significant effect from antihypertensives on ovarian cancer mortality; however, it is worth noting that 1) the sample size was too small to distinguish between beta blockers (N=141 for monotherapy), and 2) patients on antihypertensives tend to have significant risk factors for disease incidence and severity, including advanced age, higher BMI, and smoking status [PMC8585699]. Since clonidine has had some success in mitigating symptoms in patients experiencing hormonal withdrawal from bilateral oophorectomy [PMID:27249732], it may be possible for future studies to isolate whether the inclusion of this drug may impact recurrence of platinum resistant ovarian carcinoma. We have added a brief comment regarding this in the revised version of the manuscript at Lines 208-214.
Please see attachment for the full cover letter including response to all Reviewers' critiques and the revised version of the manuscript.

Reviewer 3 Report
Comments and Suggestions for Authors
The study demonstrates that 6 compounds with agonistic activity for the adrenoceptor alpha-2a enhanced cytotoxicity of carboplatin treatment.
The introduction may be expanded in terms of the relationship between the repertoire of non-redundant treatment options and prolonged patient survival.
The results of Figure 5 may be explained more in detail to describe the correlation between pharmacological activation of ADRA2A and cytotoxicity.
The specific comments: 1. The research addressed the possibility of the ADRA2A activation in promotion of chemosensitivity in ovarian cancer. Discussion on the chemosensitivity in regards to tumor microenvironment may be added to added. 2. The topic of the study is very relevant, although the role of ADRA2A in cancer is quite unknown. More detailed discussion of the role of ADRA2A in cancer drug sensitivity and potential mechanisms of drug resistance such as epithelial-mesenchymal transition may be added with citation of several references. 3. The reason why p<0.07 is significant in Figure 4 may be added.
Author Response
Please find our point-by-point responses to your comments below:
The introduction may be expanded in terms of the relationship between the repertoire of non-redundant treatment options and prolonged patient survival.
The Authors agree with the Reviewer that this could be further expanded. In the revised Introduction, we have further discussed the rationale for use of paclitaxel in combination with platinum agents as well as the use of bevacizumab and PARP inhibitors in targeting non-redundant pathways to enhance therapy (Lines 71-75 and 78-81).
The results of Figure 5 may be explained more in detail to describe the correlation between pharmacological activation of ADRA2A and cytotoxicity.
The Authors appreciate this comment from the Reviewer. We have further elaborated on the effect of ADRA2A expression in the Results section of the manuscript, Lines 158-162.
The specific comments:
- The research addressed the possibility of the ADRA2A activation in promotion of chemosensitivity in ovarian cancer. Discussion on the chemosensitivity in regards to tumor microenvironment may be added.
The Authors agree that the tumor microenvironment will be important in the clinical impact of targeting ADRA2A. Currently, the role of adrenergic signaling in the stroma is unclear, however adrenergic signaling is associated with many processes across cell types within the stroma, and indeed it appears there are isoform-dependent effects that alter tumor development and progression in a cancer-specific manner. We have revised the Discussion to highlight the potential importance of adrenergic activation in the TME to further expand on this, Lines 221-228.
- The topic of the study is very relevant, although the role of ADRA2A in cancer is quite unknown. More detailed discussion of the role of ADRA2A in cancer drug sensitivity and potential mechanisms of drug resistance such as epithelial-mesenchymal transition may be added with citation of several references.
We have expanded on the role of cAMP in epithelial-mesenchymal transition (EMT) in the Discussion, Lines 227-230. cAMP suppression downstream of G-alpha inhibition and potential impact on calcium flux downstream of G-beta/gamma may have broad effects on EMT and other hallmarks of cancer, the exact role of ADRA2A still requires mechanistic evaluation and is currently under investigation in our laboratory.
- The reason why p<0.07 is significant in Figure 4 may be added.
While we found that xylazine and dexmedetomidine had significant reduction of both viability as well as clonogenicity when combined with carboplatin, clonidine did not reach statistical significance but was only a trend. We have revised this point in the Results section to more accurately reflect the data, Lines 148-150. This is also briefly in the Discussion, Lines 172-178.
Please see attachment for the full cover letter including response to all Reviewers' critiques and the revised version of the manuscript.
